# New Insights into Adipose Tissue Macrophages in Obesity and Insulin Resistance

**DOI:** 10.3390/cells11091424

**Published:** 2022-04-22

**Authors:** Zhaohua Cai, Yijie Huang, Ben He

**Affiliations:** Heart Center, Shanghai Chest Hospital, Shanghai Jiaotong University, 241 Huaihai West Road, Shanghai 200030, China; caizhaohua@shchest.org (Z.C.); 540290682@sjtu.edu.cn (Y.H.)

**Keywords:** adipose tissue, macrophage, inflammation, obesity, insulin resistance

## Abstract

Obesity has become a worldwide epidemic that poses a severe threat to human health. Evidence suggests that many obesity comorbidities, such as type 2 diabetes mellitus, steatohepatitis, and cardiovascular diseases, are related to obesity-induced chronic low-grade inflammation. Macrophages are the primary immune cells involved in obesity-associated inflammation in both mice and humans. Intensive research over the past few years has yielded tremendous progress in our understanding of the additional roles of adipose tissue macrophages (ATMs) beyond classical M1/M2 polarization in obesity and related comorbidities. In this review, we first characterize the diverse subpopulations of ATMs in the context of obesity. Furthermore, we review the recent advance on the role of the extensive crosstalk between adipocytes and ATMs in obesity. Finally, we focus on the extended crosstalk within adipose tissue between perivascular mesenchymal cells and ATMs. Understanding the pathological mechanisms that underlie obesity will be critical for the development of new intervention strategies to prevent or treat this disease and its associated co-morbidities.

## 1. Introduction

Obesity is a metabolic disease characterized by abnormal and excessive accumulation of body fat. Obesity increases the risk of developing a wide variety of diseases including but not limited to type 2 diabetes mellitus (T2DM) and cardiovascular diseases and has been strongly associated with increased mortality [1,2]. In the past few decades, the prevalence of obesity has increased dramatically in both developing and developed nations around the world [3,4]. More recent statistics indicate that 39% of adults aged 18 years and over were overweight, and 13% were obese worldwide in 2016 [5]. Obesity is a serious public health problem with major health and economic consequences. Therefore, it is imperative that we polish our understanding of the pathological mechanisms so we can develop effective strategies to prevent or treat obesity and related comorbidities.

Insulin resistance is a key component in the etiology of T2DM, and obesity is clearly the most common cause of insulin resistance in humans [6,7]. With the ongoing worldwide obesity epidemic, there has been a parallel rise in the prevalence of T2DM [8]. It has now been widely recognized that obesity-induced chronic low-grade tissue inflammation, particularly when occurring in adipose tissue, can cause insulin resistance and T2DM [9,10]. Under obese conditions, adipose tissue undergoes a series of dynamic remodeling, including adipocyte hypertrophy, apoptosis, immune cell infiltration, extensive vascularization, and extracellular matrix remodeling [11,12]. Macrophages are the primary immune cells strongly involved in obesity-associated inflammation in both mice and humans [7,13,14,15]. Emerging studies have implicated the crosstalk between adipocytes and adipose tissue macrophages (ATMs) as critical regulators of obesity-associated inflammation and metabolic complications.

In this review, we provide an overview of the distinct subpopulations of ATMs in the context of obesity, with particular attention paid to some novel subsets of ATMs characterized by single-cell or single-nucleus RNA-sequencing (sc/snRNA-seq) technologies. We also review the latest research progress on the extensive interactions between adipocytes and ATMs mediated by miRNA-containing exosomes or mitochondria transfer. Moreover, we also describe the extended crosstalk between perivascular mesenchymal cells and ATMs. Study searches in this review were performed using the PubMed database from the National Library of Medicine.

## 2. Adipose Tissue Macrophage (ATM) Subpopulations

White adipose tissue (WAT) is comprised of a versatile group of interacting cells, including adipocytes, immune cells, and other cell types. It is evident that ATMs, the most abundant immune cells in WAT, can even represent up to 40–50% of the cells in obese adipose tissue [13]. ATMs are an extraordinarily heterogeneous population of immune cells with varied and diverse functions (as summarized in Table 1), which have been highlighted as important factors contributing to the pathogenesis of obesity and related comorbidities. Historically, ATMs have been categorized into classically activated (M1-like) and alternatively activated (M2-like) macrophages [16]. However, more and more novel ATM subpopulations have been identified [14,15,17,18,19,20,21,22,23].

ATMs were thought to be composed of two main phenotypes: classically activated macrophages and alternatively activated macrophages, which are phenotypically and functionally distinct [16]. The classically activated macrophages represent pro-inflammatory M1-like macrophages, whereas the alternatively activated macrophages are anti-inflammatory M2-like macrophages. M1-like macrophages express F4/80, CD11b, and CD11c and secrete inflammatory factors including tumor necrosis factor-α (TNF-α), interleukin-1β (IL-1β), IL-6, leukotriene B4 (LTB4), and nitric oxide (NO); whereas M2-like macrophages express F4/80, CD11b, CD301, and CD206 and exhibit increased secretion of anti-inflammatory cytokines, such as IL-4 and IL-10 [26,27]. The main roles of ATMs under lean conditions are efferocytosis of dead adipocytes, the production of anti-inflammatory cytokines, the regulation of adipocyte lipolysis, and the restriction of adipocyte progenitor proliferation [28]. Conversely, as obesity progresses, most ATMs are converted from an anti-inflammatory (M2-like) phenotype into a pro-inflammatory (M1-like) phenotype, secreting pro-inflammatory cytokines (such as TNF-α, IL-1β) and causing localized and systemic chronic low-grade inflammation, especially in WAT [29]. The insulin resistance and T2DM would progress under the influence of this inflammatory state [30,31].

ATMs can be further widely divided into adipose tissue-resident macrophages and recruited monocyte-derived macrophages. Tissue-resident macrophages are long-lived and self-renewing cells thought to have originated during embryonic hematopoiesis [32], whereas monocyte-derived macrophages are short-lived cells recruited to adipose tissues during inflammation [33]. Recently, multiple novel populations of adipose tissue-resident macrophages have been discovered in adipose depots [21,22,34]. For example, a new subpopulation of ATMs, TIM4^+^ adipose tissue-resident macrophages, has recently been revealed to play essential roles in the formation and expansion of adipose tissue during the development and diet-induced obesity [21]. Another novel resident macrophage population (sympathetic neuron-associated macrophages, SAMs) has been discovered in adipose tissue localized around neurons of the sympathetic nervous system (SNS) that mediates noradrenaline clearance and dampens SNS-to-adipocyte communications [22,34].

With the recent advances in scRNA-seq technologies, emerging evidence suggests that ATMs exhibit a wider spectrum of phenotypes and cellular identities than previously described in the context of obesity both in mice and humans. By utilizing scRNA-seq technology, Hill et al. identified three discrete ATM populations (CD11b^+^ Ly6c^+^; CD11b^+^ Ly6c^−^ CD9^+^; CD11b^+^ Ly6c^−^ CD9^−^), two of which (CD11b^+^ Ly6c^+^ and CD11b^+^ Ly6c^−^ CD9^+^) are associated with obesity [17]. CD11b^+^ Ly6c^−^ CD9^+^ ATMs reside within crown-like structures (CLS) and are lipid-laden and proinflammatory, whereas CD11b^+^ Ly6c^+^ ATMs reside outside CLSs and play angiogenic and adipogenic roles [17]. In a more recent study, Jaitin et al. provided a comprehensive single-cell adipose tissue immune atlas in mice and humans and described a novel Trem2^+^ ATM subpopulation, named lipid-associated macrophages (LAMs), in obese adipose tissue [14]. These LAMs use lipid receptor Trem2 as a sensor of extracellular lipids and play protective functions to counteract adipocyte hypertrophy, inflammation, and metabolic dysfunction [14]. Therefore, obese adipose tissue contains multiple distinct ATM populations with unique origins, tissue distributions, and functions. A comprehensive understanding of ATM heterogeneity in obesity is of great importance for the development of future therapies.

## 3. Adipocytes and ATMs Crosstalk

The crosstalk between adipocytes and macrophages in adipose tissues is crucial in obesity-induced metabolic complications. Adipocytes and macrophages can interact with each other through a variety of mechanisms, including cytokine and chemokines, microRNA-containing exosomes or microvesicles, and mitochondria transfer. These mechanisms are discussed in this section and partly summarized in Figure 1.

### 3.1. Cytokines and Chemokines as Mediators of Crosstalk

It is well established that cytokines and chemokines secreted by immune cells ignite localized and systemic inflammation, which builds up a pathogenic connection between obesity and insulin resistance. Moreover, cytokines and chemokines are the major mediators of ATM phenotype and crosstalk between adipocytes and ATMs which play critical roles in the pathogenesis of obesity and associated metabolic complications.

The first evidence for a pathophysiological link between obesity, inflammation, and insulin resistance was provided in 1901, when it was observed that the salicylate, an anti-inflammatory drug as the principal metabolite in aspirin, could beneficially control glucose in diabetics [35]. The concept was revisited, approximately a century later, when Hotamisligil et al. demonstrated that TNF-α was elevated in the adipose tissue of obese mice and neutralization of TNF-α ameliorated insulin resistance [36]. Subsequent studies reported that TNF-α-deficient mice were free from high-fat diet-induced insulin resistance [37]. Increased TNF-α production was also observed in the adipose tissue of obese humans, while the decline of TNF-α level was associated with weight loss in humans [38,39]. A milestone forward in this field was provided by two simultaneous publications that reported independently that obesity was associated with macrophage accumulation in adipose tissue, which was known as the major source of inflammatory mediators (such as TNF-α) [13,40]. Mounting studies have demonstrated that monocyte-derived macrophages are recruited into tissues (particularly into adipose tissue) via C-C chemokine receptor type 2 (CCR2) and secrete inflammatory cytokines such as TNF-α during obesity, thereby causing systemic inflammation and insulin resistance. Therefore, all these studies suggested the central role of macrophage-secreted inflammatory cytokines in the development of obesity and related comorbidities.

Many other inflammatory cytokines and mediators such as IL-1β, IL-6, monocyte chemotactic protein 1 (MCP-1), and macrophage inhibitory factor (MIF) have been implicated in the pathogenesis of insulin resistance [26,41,42,43,44,45]. The production of inflammatory cytokines during obesity has been well demonstrated to be regulated by the signaling pathway of inhibitor of κB kinase-β (IKK-β) and nuclear factor-κB (NF-κB) [46,47], c-Jun N-terminal kinase (JNK) [48,49], and the NLR family pyrin domain containing 3 (NLRP3) inflammasome [50,51,52]. There is enormous and excellent literature available on this topic [53,54,55,56]. It is worth noting that novel pathways were found to regulate the inflammatory processes. For example, Yao et al. demonstrated that the iroquois homeobox gene3 (IRX3) in ATMs functions as a novel transcriptional factor for cytokine expression and therefore accelerates the development of obesity and T2DM [57]. In addition, Fgr tyrosine kinase, which is activated by reactive oxygen species (ROS), has been highlighted as a key regulator for proinflammatory macrophages during diet-induced obesity [58].

Although dramatic progress has been made in our understanding of the role of inflammation in obesity-associated insulin resistance, clinical trials targeting inflammatory mediators of obesity and T2DM (such as TNF-α and IL-1β) to improve glycemic control in T2DM have shown limited benefits [25,59], indicating that there must be other factors contributing to decreased insulin sensitivity. Bu et al. demonstrated that growth/differentiation factor 3 (GDF3) produced from CD11c^+^ ATMs acts as a ligand of ALK7 in adipocytes to inhibit lipolysis and expand adipose tissue under obese conditions [60]. The GDF3-ALK7 signaling pathway within WAT might represent an important interactive mechanism between adipocytes and ATMs in the modulation of adiposity [23,60]. In a more recent study, Sharma et al. demonstrated that the neuroimmune guidance cue netrin-1 is essential for orchestrating macrophage fate and function that accumulate in the obese adipose tissue [61]. Myeloid-specific netrin-1 deletion reduces the ATM accumulation in adipose tissue and reprograms the ATM phenotype during obesity. Therefore, mice lacking netrin-1 in myeloid cells are protected from diet-induced obesity and metabolic dysfunction [61]. In addition, Wang et al. identified a novel cytokine Slit3 which was secreted by M2-like macrophages under cold exposure and regulated WAT beiging via stimulating norepinephrine secretion [57].

### 3.2. MicroRNA and Exosomes as Novel Mediators of Crosstalk

MicroRNAs (miRNAs) are small non-coding RNAs consisting of 19–22 nucleotides that govern gene expression at the post-transcriptional level via inhibiting the translation of target mRNA [62,63]. Mature miRNAs are formed inside the cell and perform various functions in the cytoplasm or after being released into the extracellular spaces, blood circulation, and other body fluids, (i.e., urine and saliva) [64]. Notably, miRNAs can be packaged into extracellular vesicles (EVs), including exosomes and microvesicles [65]. These EVs, cell-derived membranous structures, can deliver functional miRNAs to target cells, thereby exerting intercellular communications and interorgan crosstalk [66].

The importance of miRNAs in the immunometabolism field is increasingly recognized. Mounting evidence suggests that in addition to cytokines and chemokines, miRNAs or miRNA-containing exosomes also mediate powerful paracrine functions between ATMs and adipocytes. New mechanistic pathways involving the secretion of miRNA-containing exosomes [67,68] or microvesicles [69] by adipocytes or ATMs have been demonstrated to facilitate metabolic and inflammatory interactions between adipocytes and ATMs as well as distal target tissues, which can regulate insulin sensitivity and modulate metabolic homeostasis. Therefore, miRNAs have now been recognized as a new class of endocrine factors and are strongly implicated in the pathogenesis of obesity and related comorbidities.

Evidence is accumulating that numerous adipocyte-derived miRNAs can modulate macrophage phenotype and function and regulate insulin sensitivity via their paracrine actions. For instance, miR-155 is secreted by adipocyte-derived microvesicles from obese adipose tissue in mice. Adipocyte-derived miR-155 downregulates the protein level of suppressor of cytokine signaling 1 (SOCS1), one of its proven targets [70], and promotes pro-inflammatory M1-like macrophage polarization [69]. MiR-27a, a proven adipogenic miRNA [71,72], is significantly upregulated in the serum of obese mice and humans [73,74]. Adipocyte-derived exosomal MiR-27a facilitates macrophage activation and induces insulin resistance via inhibiting PPARγ during diet-induced obesity [73,74]. Moreover, miR-34a is elevated in the adipose tissue of obese mice and clinical patients. Adipocyte-specific depletion of miR-34a reprograms ATMs from pro-inflammatory M1-like to anti-inflammatory M2-like phenotype and protects mice from obesity-associated inflammation, glucose intolerance, insulin resistance through repressing Krüppel-like factor 4 (KLF4) expression [75]. Taken together, this evidence suggests that adipocyte-derived microvesicles or exosomes can transport miRNAs to mediate adipocyte-macrophage communications and modulate metabolic homeostasis.

ATMs can also regulate in vitro and/or in vivo insulin sensitivity by secretion of miRNA-containing exosomes. Ying et al. found that treatment of lean mice with ATM-derived exosomes harvested from obese mice causes glucose intolerance and insulin resistance, whereas treatment of obese mice with lean ATM-derived exosomes improves insulin resistance [68]. Notably, miR-155 is demonstrated to be one of the miRNAs overexpressed in obese ATM-derived exosomes which mediates the deleterious effects [68]. Moreover, it is demonstrated that mice with miR-155 knockout exhibit improved insulin sensitivity when fed with high fat diet [68]. In another work, Ying et al. treated mouse bone marrow-derived macrophages (BMDMs) with IL-4 and IL-13 to induce M2 BMDMs generation and harvested the released exosomes [76]. Treatment with M2 BMDMs-derived exosomes can improve insulin sensitivity in vivo in obese mice and in vitro (in adipocyte, myocyte, and hepatocyte) [76]. Moreover, as the authors illustrated, miR-690 is highly expressed within M2 BMDM-derived exosomes and acts as a key insulin-sensitizing miRNA which improves insulin sensitivity [76]. In addition, Liu et al. demonstrated that miR-29a is increased in ATM-derived exosomes from obese adipose tissue of mice which can be transferred into adipocytes, myocytes, and hepatocytes causing insulin resistance in vitro and in vivo [77]. These works suggest that ATM-derived exosomal miRNAs can regulate insulin sensitivity and metabolic homeostasis.

In summary, adipocytes and ATMs, residing within adipose tissue, can secrete miRNA-containing exosomes or microvesicles and elicit phenotypes via various mechanisms in obesity and associated complications. Therefore, it is promising that obesity-associated miRNAs are new therapeutic targets for obesity and related diseases.

### 3.3. Mitochondria Transfer as Novel Mediators of Crosstalk

Mitochondria are essential organelles within eukaryotic cells and play important roles in the processes such as oxidative phosphorylation, energy production, metabolic homeostasis, redox regulation, and apoptosis [78,79]. Therefore, it is not surprising that mitochondrial dysfunction is associated with many diseases, such as metabolic diseases and neurodegenerative disorders. Intercellular mitochondria transfer has attracted more and more attention within the scientific community in recent studies.

The first evidence for functional mitochondria transfer comes from the report that genetic defects in mtDNA of A549 ρ° cells can be rescued by the transfer of healthy mitochondria from human stem/progenitor cells [80]. Emerging evidence now exists suggesting that cells can release functional mitochondria that are transferred to and captured by recipient cells [81,82,83,84]. Notably, mitochondria also can be loaded in EVs and acquired by recipient cells via an EV-cell fusion manner [85,86]. Moreover, it is noted that functional mitochondria, EV-related mitochondria, and mitochondria DNA can circulate in blood both in mice and humans [84,87,88]. The intercellular mitochondrial transfer has been implicated in a variety of physiological and pathological processes, including cardiac homeostasis [89], acute lung injury repair [83], pulmonary hypertension [90], allograft rejection [87], and ischemic stroke [91].

Mitochondria transfer is emerging as a novel mediator of endogenous crosstalk between adipocytes and macrophages within adipose tissue. Brestoff et al. demonstrated that intercellular mitochondria transfer occurs in mice WAT and identified a novel adipocyte-to-macrophage mitochondria transfer axis that modulates energy homeostasis and is impaired in obesity [92]. These findings implicated a decrease in mitochondria transfer in WAT as a sign of obesity, which might be considered a promising therapeutic target [92]. It is demonstrated that intercellular mitochondria transfer is mediated by a heparan sulfates (HS)-dependent mechanism in vitro and in vivo [92]. Moreover, mice with myeloid cell-specific deletion of HS biosynthetic gene *Ext1* exhibit impaired mitochondria transfer to macrophages and increased susceptibility to diet-induced obesity [92]. In a more recent study, Rosina et al. reported that thermogenic stimuli induce the EVs released from brown adipocyte which contains damaged mitochondria [93]. These brown adipocyte-derived EVs are captured and removed by brown adipose tissue (BAT)-resident macrophages via the CD36-lysosome pathway, thereby maintaining the BAT thermogenic program [93]. Macrophage depletion causes the accumulation of mitochondria-containing EVs, suppresses the expression of mitochondrial proteins and thermogenic genes, and inhibits BAT’s thermogenic response to cold exposure [93]. These findings highlighted the brown adipocyte-to-macrophage mitochondria transfer axis as an important regulator to maintain brown adipocyte mitochondria quality control and preserve BAT homeostasis [93]. The authors even postulate that failure to remove damaged mitochondria in BAT due to impaired EV production or macrophage phagocytic activity might contribute to the progressive impairment of BAT function during obesity [93].

These findings illustrate a new pattern of immunometabolic crosstalk, adipocyte–macrophage mitochondria transfer, which contributes to maintaining systemic metabolic homeostasis. Further work is needed to elucidate how the mitochondria transfer to macrophages affects metabolism homeostasis. However, treatment targeting intercellular mitochondrial transfer may be a promising strategy for the treatment of obesity and related comorbidities.

## 4. Adipose Tissue-Resident Macrophages Directly Regulate Adiposity and Energy Storage

In addition to short-lived monocyte-derived macrophages originating from hematopoietic stem cells, there are long-lived tissue-resident macrophages that serve tissue-specific purposes [94]. However, whether adipose tissue-resident macrophages serve their tissue-specific purposes and support the function of energy storage in adipose tissue has not been completely understood.

A new subpopulation of ATMs, CCR2-independent TIM4^+^ resident macrophages, has recently been reported to modulate adiposity and energy storage in a paracrine manner via the production of platelet-derived growth factor (PDGFcc) in mice WAT [21]. Genetic deletion and pharmacological blockade of PDGFcc reduce adiposity, energy storage, and body weight, and redirect excess lipids mostly toward thermogenesis [21]. This study challenges the outdated M1/M2 macrophage polarization model in which adipose tissue-resident macrophages can serve to sense increased nutritional status and support energy storage, whereas recruited macrophages are responsible for characterizing systemic inflammation of obesity and metabolic complications. Therefore, these data strongly indicate that different developmental subsets of macrophages (including adipose tissue-resident macrophages and recruited monocyte-derived macrophages) exert different functions within adipose tissue and are independent targets of CCR2 and PDGFcc blockade. This study highlighted the additional roles of macrophages beyond classical M1/M2 polarization in obesity development and has the potential to inspire new immunomodulatory therapies that could separately manipulate energy storage and inflammation during obesity.

## 5. Sympathetic Neuron-Associated Macrophages Indirectly Affect Energy Storage

Adipose tissue is densely innervated by the sympathetic nervous system (SNS), which locally releases noradrenaline into adipose tissue and drives lipolysis and brown or beige adipocyte thermogenesis [24,95,96]. Recent advances in three-dimensional adipose tissue imaging have improved our understanding of how different cell types in adipose tissue are organized and how they interact with one another [97,98]. Moreover, the interwoven relationships between adipocytes, sympathetic nerves, and immune cells in the context of obesity have attracted widespread attention.

In 2017, two landmark studies simultaneously identified noradrenaline-degrading macrophage populations in WAT, which directly modulate the sympathetic innervation of adipocytes [22,23]. Pirzgalska et al. demonstrated that adipose tissue-residing SAMs exhibit specialized morphology for association with SNS neurons in WAT [22]. These SAMs that import and catabolize noradrenaline via noradrenaline transporter (SLC6A2) and degradation enzyme (MAMO) are dramatically increased under obese conditions [22]. Mice with genetic deletion of SLC6A2 from SAMs are resistant to obesity owing to decreased noradrenaline removal, enhanced SNS-to-adipocyte communications, and increased SNS-driven lipolysis [22]. In another study, Camell et al. found that a specialized ATM subpopulation (a SAM-like macrophage population) was activated in aged mice [34]. They further demonstrated these ATMs regulate the age-related reduction in adipocyte lipolysis in adipose tissue by degrading norepinephrine in an inflammasome-dependent manner [34]. In addition, in the more recent study by Wang et al., it is demonstrated that under cold exposure M2-like macrophages secret Slit3, which binds to ROBO1 receptor on sympathetic neurons and stimulates noradrenaline release, leading to enhanced white adipocyte beiging and thermogenesis [99]. These studies identify a sympathetic neuroimmunological role for macrophages in obesity and have opened up a whole new field of neuroimmunometabolism.

It is now well demonstrated that BAT is much more densely innervated by sympathetic nerves than WAT [100]. The triangular relationship between brown adipocytes, macrophages, and SNS has also been recognized. Wolf et al. reported that the nuclear transcription factor MECP2 is an important modulator of BAT function [101]. Mice lacking MECP2 spontaneously develop obesity due to the impairment of BAT function [101]. Further mechanistic investigation indicates that MECP2-deficient macrophage upregulates PlexinA4 expression which prevents the axonal outgrowth of Sema6A^+^ nerves and diminishes sympathetic innervation of BAT [101].

## 6. Novel View—Cross-Talk between Perivascular Mesenchymal Cells and ATMs

Although recent research mainly focuses on the role of adipocytes and macrophages in the development of metabolic adipose tissue inflammation, a new study recently highlighted that perivascular mesenchymal cells play a significant role in the regulation of chronic adipose tissue inflammation during obesity.

In the study, Shan et al. utilized scRNA-seq and identified a mouse WAT perivascular cell subpopulation, named fibro-inflammatory progenitors (FIPs) that stimulate pro-inflammatory signaling and modulate accumulation of pro-inflammatory macrophage in the adipose tissue during obesity [102]. These perivascular mesenchymal cells of the adipose tissue are critical “gatekeepers” of macrophage accumulation in obesity. It is also reported that the transcriptional regulator zinc-finger protein 423 (ZFP423) governs the inflammatory response of perivascular mesenchymal cells [102]. Using in vitro studies and in vivo mouse genetic models they determined that ZFP423 modulates NF-κB activity and that expression of ZFP423 in perivascular mesenchymal cells suppresses inflammatory signaling in FIPs and attenuates metabolic inflammation in obesity [102]. These studies highlighted an important role for perivascular mesenchymal cells in the modulation of chronic inflammation in adipose tissue during obesity, and indicate that proinflammatory perivascular mesenchymal cells are potential targets for therapeutic treatment tailored to control obesity and associated co-morbidities.

## 7. Conclusions

Mounting evidence has revealed that metabolism and immunity are inextricably interwoven. Adipocytes and ATMs are the two most important cells within adipose tissue which mediate obesity-associated inflammation and metabolic compilations. The interactions between adipocytes and ATMs have been implicated to have a key role in the development of obesity. Recently, new evidence has shown that aside from the well-known cytokines and chemokines, miRNA-containing exosomes and mitochondria transfer are also important mediators for cell–cell and organ–organ communications, especially adipocytes and macrophages interaction in obesity. Moreover, it is found that ATMs have additional roles beyond M1-M2 polarization and modulate adiposity and energy storage directly and indirectly. In addition, it is demonstrated that adipose tissue-derived perivascular mesenchymal cells are a significant “safeguard” of macrophage accrual in obesity. In a word, we now have a much deeper understanding of the sophisticated interactions between macrophages and adipocytes during obesity. This progress reveals several advances that may be exploited in treatment to suppress the inflammatory response of effector ATM and treat obesity and related complications.

## Figures and Tables

**Figure 1 cells-11-01424-f001:**
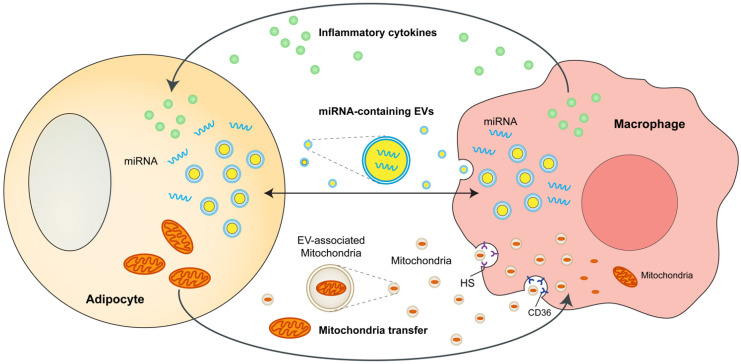
Interactions between adipocytes and ATMs in obesity. Adipocytes and macrophages interact with each other through a variety of mechanisms, including cytokine and chemokines, microRNA-containing exosomes or microvesicles, and mitochondrial transfer. ATMs, adipose tissue macrophages; miRNA, microRNA; EVs, extracellular vesicles; HS, heparan sulfates.

**Table 1 cells-11-01424-t001:** Adipose tissue macrophage (ATM) subpopulations.

Macrophage Subpopulation	Characteristics	Function
M1-like (classically activated) [16]	F4/80^+^, CD11b^+^, CD11c^+^	Pro-inflammatory phenotype that secrete inflammatory factors including TNF-α, IL-1β, IL-6, and NO
M2-like (alternatively activated) [16]	F4/80^+^, CD11b^+^, CD301^+^, CD206^+^	Anti-inflammatory phenotype that secrete anti-inflammatory cytokines, such as IL-4 and IL-10
TIM4^+^ Adipose tissue-resident Macrophages [21]	F4/80^+^, CD11b^+^, TIM4^+^, CD11c^−^; expressing PDGFcc	Tissue-resident macrophages that modulate adipocyte size and lipid storage
Sympathetic neuron-associated macrophages [24,25]	expressing the NE transporter *Slc6a2* and the NE degradation enzyme MAOA	A novel resident macrophage subpopulation that mediates noradrenaline clearance and dampens SNS-to-adipocyte communication
CD9^+^ ATM [17]	CD11b^+^, Ly6c^−^, CD9^+^; residing within CLS	Pro-inflammatory subpopulation
Lipid-associated macrophages [14]	CD9^+^, CD63^+^, Trem2^+^	Tissue-resident macrophages that counteract inflammation and adipocyte hypertrophy

Tumor necrosis factor-α (TNF-α), interleukin-1β (IL-1β), interleukin-6 (IL-6), nitric oxide (NO), interleukin-4 (IL-4), interleukin-10 (IL-10), platelet-derived growth factor (PDGFcc), norepinephrine (NE), solute carrier family 6 member 2 (*Slc6a2*), monoamine oxidase A (MAOA), sympathetic nervous system (SNS), crown-like structure (CLS).

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
