# Peer review of "New Insights into Adipose Tissue Macrophages in Obesity and Insulin Resistance"

_cells, 2022, doi:10.3390/cells11091424_

Round 1

Reviewer 1 Report

This reviewer is usually a rather critical supporter of careful evaluation as to whether a manuscript really demonstrates data that proof the appropriate hypotheses answers the questions posed. This job is important as there are numerous submissions that fail to be convincing, have poor designs, show flaws in methods, and most importantly have conclusion too farfetched. Most articles I do review , I reject due to the above mentioned reasons.

Having said that I must say that I am stunned by the fact that I do not find major issues for improvements in this nice review. It is of actual timing; the design is well-structured and the content is concisely presented. This refers also to the figure and table. I believe that this review warrants the publication in “Cells”.

I really would like to encourage the authors to generate another creative figure that either contains a perspective to that research might want to turn to or that comprises the different mechanisms active in macrophages promoting or protecting from diet induced obesity. I believe that these authors might be able to pull something creative off. But that request is optional.

Author Response

We thank the reviewer so much for such positive comments for our review article. Due to the deadline for revision, we don’t have enough time for the figure as suggested, but we will keep this wonderful suggestion in mind.

Reviewer 2 Report

The review, entitled “New insights into Adipose tissue macrophages in Obesity and Insulin resistance”, tried to summary adipose tissue-specific macrophages via 1) characterizing the diverse subpopulations of ATMs in the context of obesity; 2) reviewing the recent advance on the role of the extensive crosstalk between adipocytes and ATMs in obesity; and 3) focusing on the extended crosstalk within adipose tissue between perivascular mesenchymal cells and ATMs. The goal of the review is to understand the pathological mechanisms for the development of new intervention strategies to prevent and/or treat related disease as well as associated comorbidities. It is common that adipose tissue macrophages are the tissue resident macrophages and help maintain tissue homeostasis in steady state. The main function of ATMs is to engulf dead adipocytes to help in the cellular turnover and plays in obesity-associated inflammation and metabolic diseases by involved in lipid and energy metabolism and mitochondrial function in adipocytes. After reading the manuscript thoroughly, I have some concerns and hope to see that authors address them in the revised version.

  1. It is 52% plagiarism by checking ithenticated.com. I do think it is too high to be publication in the higher journal like “Cells”.
  2. The title means “New insight”, but the incited references regarding ATMs are not published presently. Please read more recently published articles either research articles or reviews.
  3. Important concluding sentence such as “Macrophages are the primary immune cells involved in obesity associated inflammation in both mice and humans” in line 42-43 of page 1 is needed to have ref.
  4. Table 1 needs to be provided the refences also.
  5. It is known that macrophage can engulf many from small molecular to deed cells. However, substances that enter the cells are strictly limited and selected. So, the Figure 1 needs to be modified how the miRNAs-containing EVs and macrophage mitochondria enter the adipocyte? It is exactly new insights.
  6. I don’t know what research Dr. He’ lab is doing. I strongly suggest that authors add their own related works either published or upgoing.

Author Response

  1. It is 52% plagiarism by checking ithenticated.com. I do think it is too high to be publication in the higher journal like “Cells”.

We rewrote almost the whole of the manuscript. We have tried our best to polish the writing and include more recent findings of the topic in our review article.

  1. The title means “New insight”, but the incited references regarding ATMs are not published presently. Please read more recently published articles either research articles or reviews.

According to the reviewer’s suggestion, we have read more recently published articles and cited them in the new version of the manuscript. Please find some of the new articles that we have cited:

The Journal of clinical investigation. 2019;129:4032-4040;

Cell metabolism. 2018;28:300-309 e304;

Cell metabolism. 2021;33:437-453 e435;

FASEB journal. 2021;35:e21417;

Nature metabolism. 2020;2:974-988;

Immunometabolism. 2019;1(2):e190010.

Cell. 2020;183:94-109 e123;

Cell metabolism. 2022;34:533-548 e512;

Cell metabolism. 2017;26:686-692 e683;

Cell metabolism. 2018;27:226-236 e223;

Nature. 2019;569:229-235

  1. Important concluding sentence such as “Macrophages are the primary immune cells involved in obesity associated inflammation in both mice and humans” in line 42-43 of page 1 is needed to have ref.

According to the reviewer’s suggestion, we have added references 7, 13, 14, 15 for concluding sentence “Macrophages are the primary immune cells involved in obesity associated inflammation in both mice and humans” (Line 40-41 in page 1 of the new version of the manuscript).

  1. Table 1 needs to be provided the refences also.

According to the reviewer’s suggestion, we have added references for Table 1.

  1. It is known that macrophage can engulf many from small molecular to deed cells. However, substances that enter the cells are strictly limited and selected. So, the Figure 1 needs to be modified how the miRNAs-containing EVs and macrophage mitochondria enter the adipocyte? It is exactly new insights.

According to the reviewer’s suggestion, we have modified Figure 1.

As it is reported in Cell metabolism. 2022;34:533-548 e512 & Cell metabolism. 2021;33:270-282 e278, mitochondria-containing EVs were internalized by adipose tissue macrophages in CD36- or heparan sulfates-dependent manners and eliminated by lysosome. Therefore, we have added some information in Figure 1 to show that  miRNAs-containing and mitochondria-containing EVs were internalized via endocytosis.

  1. I don’t know what research Dr. He’ lab is doing. I strongly suggest that authors add their own related works either published or upgoing.

We agree with the reviewer that it is perfect to add our own related work in this review. Previously, our lab (Dr. He’s Lab) mainly focused on the molecular mechanisms underlying cardiovascular diseases. Recently, we pay more attention to adipose tissue and its role in obesity and cardiovascular complications. We apologized that we haven’t had any published work that can be added in this review. However, we believe that we will have some promising findings in the following years.

Reviewer 3 Report

The manuscript is novel and presents an updated view regarding the role of macrophages in adipose tissue in obesity and insulin resistance. The structure of the manuscript is adequate. All points discussed are consistent with the purpose of the text. However, I have the following comments.

I. Major Comments.
1. Improve the writing of the manuscript. It is possible to read the text, but an edition is required. Especially in the section "Cytokines and chemokines as mediators of crosstalk".

2. After the introduction, I suggest including a section on the methodology for writing the manuscript (criteria for selecting cited manuscripts). This change is not difficult, and will help improve the scientific quality of the manuscript. Example: https://doi.org/10.3390/nu13103384

3. In the condition of obesity in adipose tissue an increase in oxidative stress and inflammatory response is observed. In this regard, the increase in the activity of the transcription factor NF-kB and an alteration in the activity of the transcription factor PPAR-gamma. I suggest briefly discussing this point. doi: 10.1039/d0fo01790f

II. Minor comments:
1. Improve the wording of the objective of the manuscript.
2. Figure legend 1. Define HS.

Author Response

Review 3:

Major Comments.

  1. Improve the writing of the manuscript. It is possible to read the text, but an edition is required. Especially in the section "Cytokines and chemokines as mediators of crosstalk".

According to the reviewer’s suggestion, we have tried our best to polishe the writing of the whole manuscript.

  1. After the introduction, I suggest including a section on the methodology for writing the manuscript (criteria for selecting cited manuscripts). This change is not difficult, and will help improve the scientific quality of the manuscript. Example: https://doi.org/10.3390/nu13103384

Following the reviewer’s suggestion, we have included a sentence to describe how we find the cited manuscript.  We added the sentence “Study searches in this review were performed using the PubMed database from the National Library of Medicine” at the end of the Introduction part.

  1. In the condition of obesity in adipose tissue an increase in oxidative stress and inflammatory response is observed. In this regard, the increase in the activity of the transcription factor NF-kB and an alteration in the activity of the transcription factor PPAR-gamma. I suggest briefly discussing this point. doi: 10.1039/d0fo01790f

We agree with the reviewer’s suggestion. We have added a novel finding that “Fgr tyrosine kinase, which is activated by reactive oxygen species (ROS), has been highlighted as a key regulator for proinflammatory macrophage during diet-induced obesity” in line 335-337 in page 5 of the new version.

Minor comments:

  1. Improve the wording of the objective of the manuscript.

Following the reviewer’s suggestion, we rewrote the objective of the manuscript. Please see it: “In this review, we provide an overview of the distinct subpopulations of ATMs in the context of obesity, with particular attention paid to some novel subsets of ATMs characterized by single-cell or single-nucleus RNA-sequencing (sc/snRNA-seq) technologies. We also review the latest research progress on the extensive interactions between adipocytes and ATMs mediated by miRNA-containing exosomes or mitochondria transfer. Moreover, we also describe the extended crosstalk between perivascular mesenchymal cells and ATMs”. (As shown in Page 1-2 in the new version of the manuscript).

  1. Figure legend 1. Define HS.

Following the reviewer’s suggestion, we have defined HS in the figure legend.

Round 2

Reviewer 2 Report

very appreciated for the authors'revised version, which addressed my concerns. I recommend current version is ready to be published in "Cells" 

Reviewer 3 Report

The authors answered all my comments. Manuscript was improved. Therefore, manuscript can be accepted in the present form.